# Longitudinal Analysis of Placental *IRS1* DNA Methylation and Childhood Obesity

**DOI:** 10.3390/ijms26073141

**Published:** 2025-03-28

**Authors:** Ariadna Gómez-Vilarrubla, Maria Niubó-Pallàs, Berta Mas-Parés, Alexandra Bonmatí-Santané, Jose-Maria Martínez-Calcerrada, Beatriz López, Aaron Peñas-Cruz, Francis de Zegher, Lourdes Ibáñez, Abel López-Bermejo, Judit Bassols

**Affiliations:** 1Maternal-Fetal Metabolic Research Group, Girona Institute for Biomedical Research (IDIBGI), 17190 Salt, Spain; agomez@idibgi.org (A.G.-V.); mniubo@idibgi.org (M.N.-P.);; 2Pediatric Endocrinology Research Group, Girona Institute for Biomedical Research (IDIBGI), 17190 Salt, Spain; 3Department of Gynecology, Dr. Josep Trueta Hospital, 17007 Girona, Spain; 4Control Engineering and Intelligent Systems (eXiT), University of Girona, 17003 Girona, Spain; 5Department of Development and Regeneration, University of Leuven, 3000 Leuven, Belgium; 6Endocrinology, Pediatric Research Institute, Sant Joan de Déu Children’s Hospital, 08950 Esplugues de Llobregat, Spain; 7Centro de Investigación Biomédica en Red de Diabetes y Enfermedades Metabólicas Asociadas, Instituto de Salud Carlos III, 28029 Madrid, Spain; 8Department of Pediatrics, Dr. Josep Trueta Hospital, 17007 Girona, Spain; 9Department of Medical Sciences, University of Girona, 17003 Girona, Spain

**Keywords:** childhood obesity, DNA methylation, fetal programming, placenta, leukocytes, metabolic risk

## Abstract

Accumulating evidence suggests that the predisposition to metabolic diseases is established in utero through epigenomic modifications. However, it remains unclear whether childhood obesity results from preexisting epigenomic alterations or whether obesity itself induces changes in the epigenome. This study aimed to identify DNA methylation marks in the placenta associated with obesity-related outcomes in children at age 6 and to assess these marks in blood samples at age 6 and whether they correlate with obesity-related outcomes at that time. Using an epigenome-wide DNA methylation microarray on 24 placental samples, we identified differentially methylated CpGs (DMCs) associated with offspring BMI-SDS at 6 years. Individual DMCs were validated in 147 additional placental and leukocyte samples from children at 6 years of age. The methylation and/or gene expression of *IRS1* in both placenta and offspring leukocytes were significantly associated with various metabolic risk parameters at age 6 (all *p* ≤ 0.05). Logistic regression models (LRM) and machine learning (ML) models indicated that *IRS1* methylation in the placenta could strongly predict offspring obesity. Our results suggest that *IRS1* may serve as a potential biomarker for the prediction of obesity and metabolic risk in children.

## 1. Introduction

Childhood obesity is a major public health concern, affecting millions of children and posing significant risks for chronic cardiometabolic disorders traditionally associated with adulthood [1,2]. While genetic and environmental factors contribute to obesity, their predictive power remains limited, underscoring the potential role of epigenetics in understanding the obesity risk [3,4].

DNA methylation, the most widely studied epigenetic mechanism, has been linked to childhood obesity in numerous studies [5]. Candidate-based approaches have identified obesity-associated genes in infants and children [6,7,8], while epigenome-wide association studies (EWAS) have revealed hundreds of differentially methylated sites [9,10]. However, many of these studies lack robust adjustment for confounding factors, and the findings are rarely replicated.

Emerging evidence suggests a bidirectional relationship between obesity and epigenetic changes. Some studies propose that excess weight alters the epigenome, while others indicate that epigenetic changes may predispose individuals to obesity [11,12]. Longitudinal studies are essential to clarify this causal relationship, yet few have examined DNA methylation at multiple time points in children. For instance, a longitudinal study using data from the ALSPAC cohort investigated the associations between early postnatal weight gain and DNA methylation in cord blood, as well as in peripheral blood samples taken at ages 7 and 17 [13]. A recent prospective cohort analysis including 31 children found that the DNA methylation signatures in cord blood remained stable in the saliva at age 6–12 years [14]. However, evidence of the persistence of DNA methylation marks over time is scarce.

Although epigenetic patterns vary across cells [15], some studies have shown that epigenetic marks can be consistent across tissues [16] and over time [17]. Notably, blood-based epigenetic biomarkers may reflect signatures in biologically relevant tissues, such as the adipose tissue [18]. Among these, key regulators of insulin signaling and metabolism may be particularly relevant in the context of the obesity risk. Insulin receptor substrate 1 (IRS1), a crucial mediator of insulin action, plays a central role in glucose homeostasis, lipid metabolism, and the energy balance [19]. Given the placenta’s essential function in fetal nutrient supply and metabolic programming, epigenetic modifications of *IRS1* in placental tissue could contribute to long-term metabolic adaptations in offspring. In this context, we hypothesize that DNA methylation marks in placental tissue at birth may influence the body composition and adiposity in offspring at the age of 6 years. We further propose that, while some methylation marks may be unstable and tissue-specific, others could persist over time and serve as early predictors of obesity and metabolic risk in children.

Specifically, we aimed (1) to identify DNA methylation marks in the placenta associated with obesity-related outcomes in offspring at age 6 and (2) to assess these marks in blood samples at age 6 and to study their correlation with obesity-related outcomes at that time.

## 2. Results

### 2.1. Participants Characteristics

Table 1 presents the characteristics of the mothers and their offspring in the study population. All women were apparently healthy and had no known diseases except for possible overweight (mean pregestational BMI: 25 kg/m^2^). The children in both groups had similar anthropometric characteristics.

For study purposes, the subjects in the validation group were separated by sex (Appendix A). No differences were observed between boys and girls apart from the body mass distribution, as boys had a lower fat mass (FM) and higher lean body mass (LBM) compared to girls.

### 2.2. Placental DMCs Associated with Obesity Risk in Offspring

A total of 977 CpGs, which were annotated to 816 genes, were differentially methylated in association with BMI-SDS in offspring at 6 years of age. From these, 538 CpGs (55%) presented positive associations (hypermethylated) and 439 CpGs (45%) presented negative associations (hypomethylated). A compilation of the top hypermethylated and hypomethylated DMCs is shown in Table 2, and the full list is shown in Appendix A. Among the top hypermethylated DMCs, several were observed to be annotated to the same gene, align in the same direction, and be located close to each other (e.g., *TMEM218*, *ASPG*, and *IRS1*), suggesting they might have greater biological significance.

The genomic distribution of the DMCs with respect to the CpG island and gene regions is shown in Figure 1. We observed the enrichment of DMCs within the CpG island, mainly corresponding to hypermethylated CpGs. With respect to gene regions, we observed the enrichment of hypomethylated CpGs within the body, and the hypermethylated CpGs were mainly distributed in TSS1500, TSS200, and the gene first exon.

The gene set enrichment (KEGG pathways) analysis revealed that DMCs associated with BMI-SDS in offspring were significantly enriched in pathways related to cell proliferation, survival, and metabolism. These pathways included signal transduction (mTOR signaling pathway, phosphatidylinositol signaling system, MAPK signaling pathway, AMPK signaling pathway), endocrine and metabolic diseases (type 2 diabetes, insulin resistance, non-alcoholic fatty liver disease), neurodegenerative diseases (spinocerebellar ataxia, Alzheimer’s disease), carbohydrate metabolism (inositol phosphate metabolism), lipid metabolism (biosynthesis of unsaturated fatty acids), and cancer-related pathways (Appendix A).

The clustergram of the top pathways and overlapped genes (Appendix A) showed that phosphatidylinositol 3-kinase (*PIK3*) regulatory subunits 1 and 2, protein kinase C beta (*PRKCB*), insulin receptor substrate 1 (*IRS1*), and insulin receptor (*INSR*), which were significantly enriched in the KEGG pathways analysis (Appendix A), were common genes across the top pathways, highlighting their significant roles in metabolic regulation.

Given that the *IRS1* gene had four hypermethylated DMCs (two CpGs among the top hypermethylated ones and two additional CpGs located nearby), and was one of the most abundant genes present in the top enriched pathways, it was selected to be validated in an extended number of samples (validation group).

### 2.3. Placental IRS1 Methylation

The methylation analysis by pyrosequencing confirmed that, among the four studied CpGs annotated to *IRS1*, CpG2 was related to an increased BMI in offspring, as its methylation in the placenta was significantly higher in children with BMI-SDS > p50 compared to those with BMI-SDS < p50 (*p* = 0.01) (Appendix A).

Bivariate correlations showed that placental *IRS1* (CpG2) methylation was associated with several parameters related to the body composition and metabolic risk in offspring at 6 years of age, including weight-SDS, BMI-SDS, Δ BW-SDS to weight-SDS, waist, hip, waist-to-height ratio, LBM, FM, and subcutaneous and preperitoneal fat (all *p* ≤ 0.05) (Table 3).

When separating the population by sex, nearly all correlations were observed in both boys and girls (all *p* ≤ 0.05) (Table 3), who displayed comparable levels of *IRS1* (CpG2) methylation (Appendix A). Moreover, in girls, *IRS1* (CpG2) methylation also correlated with insulin (*p* = 0.03) (Table 3 and Appendix A). Most of these correlations maintained statistical significance in multiple regression analyses (MRA) after adjusting for potential confounding variables (bold *p*-values in Table 3).

### 2.4. Placental IRS1 Expression

Placental *IRS1* expression was positively correlated with placental *IRS1* (CpG2) methylation (r = 0.170, *p* = 0.04) (Appendix A).

In turn, placental *IRS1* expression was correlated with several parameters related to the body composition and metabolic risk in offspring at 6 years of age, including LBM, visceral fat, insulin, and HOMA-IR (all *p* ≤ 0.05) (Table 3). The correlations between placental *IRS1* expression and visceral fat remained significant after adjusting for potential confounding variables in the MRA.

When separating the population by sex, no differences were observed in *IRS1* expression between boys and girls Appendix A). Boys showed positive correlations between placental *IRS1* expression and parameters related to body composition (weight-SDS, hip circumference, LBM, and visceral fat), while girls exhibited positive correlations with markers of insulin resistance (insulin and HOMA-IR) (all *p* ≤ 0.05). After the MRA, the correlation between placental *IRS1* expression and visceral fat remained significant in boys, while the correlation with insulin and HOMA-IR was maintained in girls (Table 3 and Appendix A).

### 2.5. Leukocyte IRS1 Methylation and Expression

*IRS1* (CpG2) methylation failed to show significant correlations with *IRS1* expression in offspring leukocytes, nor did it correlate with placental *IRS1* methylation. Additionally, no significant correlations were observed with parameters related to the body composition and metabolic risk in offspring 

However, leukocyte *IRS1* expression was correlated with markers of metabolic risk in offspring at 6 years of age, including the waist-to-height ratio and visceral fat, across the entire population. These correlations maintained statistical significance after adjusting for potential confounding variables in the MRA (all *p* ≤ 0.05) (Table 4). These correlations were stronger and more prevalent in girls, who showed significant correlations with weight-SDS, BMI-SDS, Δ BW-SDS to weight-SDS, waist, waist-to-height ratio, FBM, subcutaneous fat, visceral fat, insulin, and HOMA IR (all *p* ≤ 0.05) (Table 4 and Appendix A). Most of these correlations remained statistically significant in the MRA after adjusting for potential confounding variables. No significant correlations were observed in boys, and no significant differences in *IRS1* (CpG2) methylation were found between boys and girls (Appendix A).

### 2.6. Prediction of Obesity-Related Parameters

LRMs were developed with children’s BMI-SDS status (children with higher BMI-SDS [BMI-SDS > p50] or lower BMI-SDS [BMI-SDS < p50]) as the outcome variable. Predictors included placental *IRS1* (CpG2) methylation, age, and sex. *IRS1* (CpG2) methylation significantly predicted childhood BMI-SDS (*p* = 0.01), explaining 7.6% of the variance (R^2^ = 0.076). For each 1% increase in methylation, the odds of a higher BMI-SDS increased by 2.32 (Appendix A). A similar model predicting the visceral fat status (higher visceral fat-SDS [>p50] or lower visceral fat-SDS [<p50]) used placental and leukocyte *IRS1* expression as predictors, along with age and sex. Both were significant (*p* = 0.03 and *p* = 0.02), explaining 7.0% and9.6% of the variance, respectively. For each 1% increase in *IRS1* expression, the odds of higher visceral fat increased by 2.69 (placenta) and 5.23 (leukocytes).

ML models incorporating prenatal and infancy clinical variables also showed that placental *IRS1* (CpG2) methylation was key in predicting the BMI, alongside the maternal blood pressure, gestational smoking, children’s weight at 6 months, gestational age, maternal BMI, paternal obesity, and breastfeeding duration. The model achieved recall of 0.73, precision of 0.78, accuracy of 0.76, and an F1 score of 0.75 (Figure 2).

## 3. Discussion

This longitudinal study investigated early-life epigenetic marks as predictors of childhood obesity, identifying 977 DMCs in the placenta associated with children’s BMI-SDS at 6 years of age. Among these, *IRS1* emerged as one of the most hypermethylated DMCs. Further analysis of *IRS1* methylation and expression in the placenta and blood at 6 years revealed significant associations with metabolic risk and obesity-related parameters at this age. Notably, *IRS1* expression in leukocytes at 6 years was also linked to children’s metabolic risk parameters. Prediction models, employing both LRM and ML approaches, suggested that placental *IRS1* methylation is a potential robust predictor of the childhood obesity risk.

Among the 977 DMCs in the placenta associated with children’s BMI-SDS at 6 years of age, both hyper- and hypomethylated sites were identified. Notably, several of the top hypermethylated DMCs were annotated to the same genes—*TMEM218*, *ASPG*, and *IRS1*—and aligned in the same direction, clustering closely together, suggesting a potentially greater biological effect. Most hypermethylated DMCs were located in CpG islands, whereas hypomethylated DMCs were also found in shore regions. This distribution aligns with evidence linking CpG island methylation to stable gene silencing and disease processes, while methylation in shore regions is known to display highly conserved, tissue-specific patterns [20,21].

The pathway analysis revealed that placental DMCs associated with childhood obesity were involved in cell proliferation, survival, and metabolism. Type 2 diabetes and the mammalian target of rapamycin (mTOR) signaling pathway ranked highest, with *IRS1* among the overlapping genes. mTOR regulates insulin signaling via *IRS1* in metabolic tissues and is implicated in diseases like type 2 diabetes, obesity, and cancer [22]. An EWAS in blood from children with obesity and children of normal weight found that “*IRS1* target genes” were among the top enriched pathways in the identified CpGs [23]. These findings underscore the importance of *IRS1*, whose methylation and expression were longitudinally analyzed in placental samples and children’s blood at age 6 in the present study.

The validation analysis in placental samples revealed a significant relationship between *IRS1* (CpG2) methylation and childhood BMI-SDS, consistent with the EWAS results. Placental *IRS1* methylation and expression were positively correlated. Although *IRS1* (CpG2) is located within a CpG island, where gene silencing is typically expected, it is located in the 3’UTR rather than the promoter region. Similar positive associations between 3’UTR methylation and gene expression have been reported in cancer, emphasizing the 3’UTR’s epigenetic significance and potential as a disease biomarker [24].

IRS1 is a key component of insulin signaling [19], binding to the phosphorylated insulin receptor to activate downstream cascades [25]. *IRS1* knockout mice suggest its role in adipocyte differentiation [26]. A nearby SNP has been linked to increased visceral fat, insulin resistance, dyslipidemia, diabetes, and coronary artery disease risks [27]. *IRS1* expression in human adipose tissue relates to the fat distribution and metabolic traits [28]. Our findings were consistent with the aforementioned results, showing that *IRS1* CpG2 methylation and expression in the placenta correlated with children’s metabolic parameters at age 6, including obesity (waist-to-height ratio and subcutaneous, preperitoneal, and visceral fat) and insulin resistance markers (insulin and HOMA-IR).

Some of the correlations showed sexual dimorphism. Boys presented stronger correlations between placental *IRS1* expression and fat distribution, while placental *IRS1* expression in girls strongly correlated with insulin resistance-related variables. Sexual dimorphism has previously been reported as the genetic basis of fat distribution [29], and sex-dimorphic effects on fasting insulin at *IRS1* loci have also been described [30]. Recent studies have also shown sex differences in subcutaneous adipose tissue *IRS1* mRNA expression in adults with obesity [31]. Our results suggest that this sexual dimorphism may already be present before birth and influence visceral fat accumulation in boys by age 6.

It is worth noting that methylation marks are often tissue-specific, and placental epigenetic marks do not necessarily correspond to systemic methylation changes. In this regard, the lack of correlation between *IRS1* methylation in the placenta and in leukocytes could be attributed to tissue-specific variability in DNA methylation patterns [20]. However, we did observe an association between *IRS1* methylation and/or expression and children’s metabolic parameters at two time points (placenta at birth and leukocytes at 6 years), highlighting the potential significance of this gene in obesity development across different life stages. Finally, prediction models using both LRM and ML methods demonstrated that *IRS1* methylation in the placenta strongly predicts children’s obesity.

Our data provide evidence that placental *IRS1* methylation could potentially serve as a biomarker for the childhood obesity risk. This study highlights the potential of early-life epigenetic marks, such as those identified in *IRS1*, as predictive tools in identifying children at a higher risk of developing obesity and related metabolic conditions. This could enable targeted early interventions aimed at modulating risk factors and improving long-term health outcomes.

The major strengths of our study include the longitudinal design and the availability of placental and children’s blood samples from a large, well-phenotyped cohort. Additionally, placenta-based biomarkers may offer clinical relevance due to the simplicity of obtaining and analyzing samples, making them promising candidates for predictive biomarkers. However, some limitations must be acknowledged. The small sample size prevented stratification based on BMI-SDS categories (BMI-SDS < 1, normal weight; 1 < BMI-SDS < 2, overweight; and BMI-SDS > 2, obese). Moreover, the imbalance between children with obesity and children without obesity at 6 years required us to define the BMI status using the 50th percentile as a reference. A larger sample would improve the statistical power and ML model reliability, leading to more robust conclusions.

Moreover, it would be valuable to validate these results in other independent cohorts, as well as to investigate the applicability of our findings to other ethnic groups.

Another limitation of this study was the inability to include additional variables such as the maternal diet, socioeconomic status, and paternal obesity in the multiple regression analysis, as data on these factors were either unavailable or incomplete.

In conclusion, we present the first longitudinal data on *IRS1* methylation and expression in the placenta and children’s leukocytes at age 6. We propose that placental *IRS1* methylation may play a role regulating obesity and metabolic risk parameters and could serve as an early biomarker for metabolic risk. Our findings highlight the importance of altered placental DNA methylation in fetal programming and the development of non-communicable diseases like childhood obesity. This paves the way for the future use of placenta-based epigenetic biomarkers to predict disease and metabolic dysfunction.

## 4. Materials and Methods

### 4.1. Study Participants

This study included 171 pregnant women and their infants from a population-based prenatal cohort in Girona. Of these, 24 women were part of the screening analysis to identify differentially methylated CpG sites (DMCs) associated with BMI-SDS in offspring at age 6 using an EWAS. The remaining 147 mother–infant pairs formed the validation group, used to validate 4 selected DMCs related to the *IRS1* gene through pyrosequencing. The sampling method ensured the accurate representation of maternal (pre-pregnancy BMI and age) and offspring characteristics (gender and BMI at 6 years) (Table 1). The study design flow chart is shown in Figure 3, with the inclusion and exclusion criteria detailed in the Appendix A.

### 4.2. Biological Samples

Four placental tissue biopsies and fasting peripheral blood samples from children were collected at birth and at 6 years of age, respectively. Further details of sample collection and handling are provided in the Appendix A.

### 4.3. Clinical Assessments

Information on pregnancy, labor, and delivery characteristics was obtained from standardized medical records. The maternal weight at the beginning of gestation (6–9 weeks) was used as a proxy for the pregestational weight. The pregestational BMI was calculated as the pregestational weight divided by the height squared (kg/m^2^). Newborns were weighed and measured immediately after delivery using a calibrated scale and measuring board, respectively. Gestational age- and sex- adjusted z-scores (SDS) for birth weight and length were calculated using regional norms [32].

Anthropometric measurements and blood samples from children were collected during a follow-up visit at age 6. Further details are provided in the Appendix A.

### 4.4. Infinium MethylationEPIC BeadChip Microarray

A placental epigenome-wide DNA methylation microarray was performed in 24 samples (screening group) using the Infinium^®^ Human MethylationEPIC BeadChip (Illumina, San Diego, CA, USA), covering a total of 850,000 CpGs. Placental DNA was isolated using the Gentra Pure-Gene Tissue Kit (QIAGEN, Hilden, Germany). DNA quality checks, bisulfite treatment, and hybridization were performed at the Epigenomics Unit IIS La Fe (Valencia, Spain), following the manufacturer’s protocol (Illumina Infinium HD methylation protocol).

Further details of data normalization and the statistical analysis of associations between DMCs and BMI-SDS in offspring are provided in the Appendix A.

Gene annotation for the DMCs associated with offspring BMI-SDS was performed as previously described elsewhere [33]. Raw DNA methylation data have been deposited in the Gene Expression Omnibus data repository under accession number GSE192812.

### 4.5. Pathway Analysis

Gene set enrichment analysis was conducted using the Enrichr analysis tool version 3.3 (Ma’ayan Lab at the Icahn School of Medicine at Mount Sinai, New York, NY, USA). Enrichr identified relevant Gene Ontology (GO) terms and functional pathways associated with the genes annotated for the DMCs in relation to offspring BMI-SDS.

### 4.6. DNA Methylation Assessment

The methylation percentages of the four DMCs, derived from the microarray analysis and annotated to the *IRS1* gene, were assessed in the placental and blood samples from the validation group using sodium bisulfite pyrosequencing. Amplifying primers were designed with the PyroMark Assay Design 2.0 software (QIAGEN, Hilde, Germany) and are listed in Appendix A. Details of DNA isolation, bisulfite conversion, real-time PCR amplification, and pyrosequencing are provided in the Appendix A.

### 4.7. Gene Expression Assessment

*IRS1* expression levels were assessed by quantitative reverse transcription PCR (RT-qPCR). Further details are provided in the Appendix A.

### 4.8. Statistical Analysis

The pyrosequencing and gene expression data from the validation group were analyzed using SPSS version 22.0 (IBM, New York, NY, USA). Further details of the data analysis and statistical tools are provided in the Appendix A.

### 4.9. Prediction Analysis

Logistic regression models (LRM) were used to examine the contributions of the selected DMCs to the prediction of obesity and metabolic risk. The case–control status (higher or lower risk) served as the outcome variable, while the methylation or expression levels, along with age and sex, were predictors. Further details of the predictive models using machine learning (ML) analyses are provided in Appendix A.

The study was approved by the Ethics Committee of the Institutional Review Board of Dr. Josep Trueta Hospital of Girona (Spain). Informed written consent was obtained from all participants, and data were anonymized. Data will be made available upon reasonable request.

## Figures and Tables

**Figure 1 ijms-26-03141-f001:**
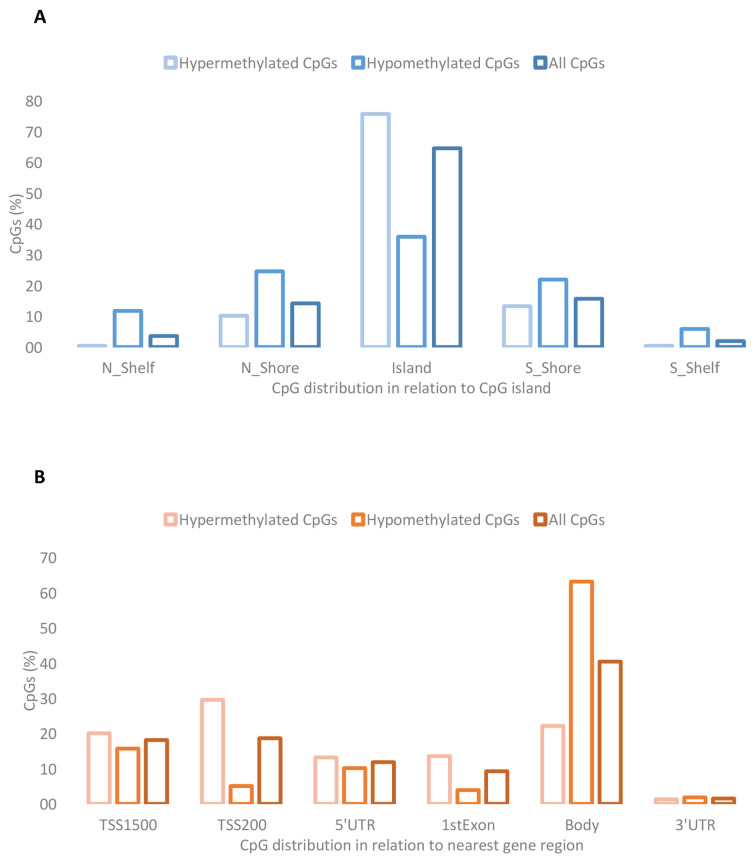
Distribution of hypermethylated and hypomethylated DMCs in relation to (**A**) CpG island regions and (**B**) the nearest gene region.

**Figure 2 ijms-26-03141-f002:**
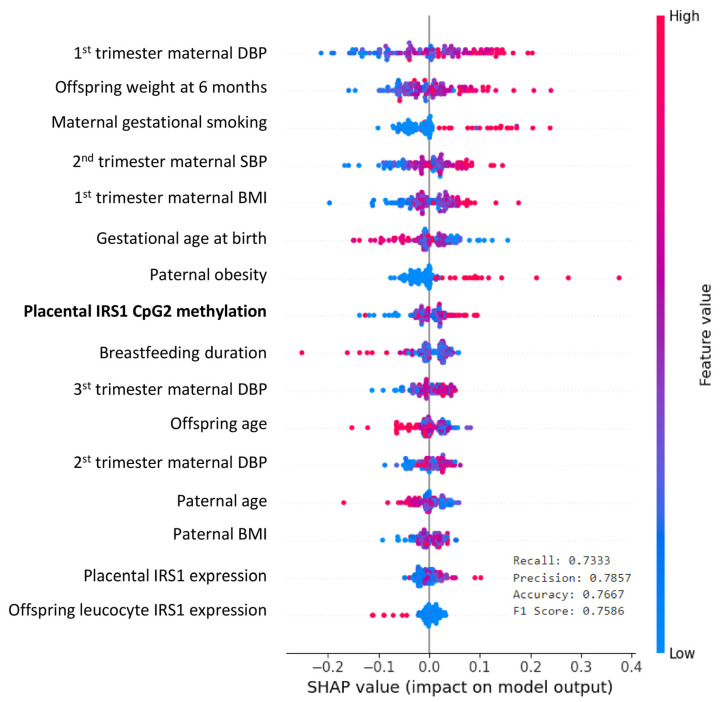
Machine learning model to predict obesity risk in 6-year-old children.

**Figure 3 ijms-26-03141-f003:**
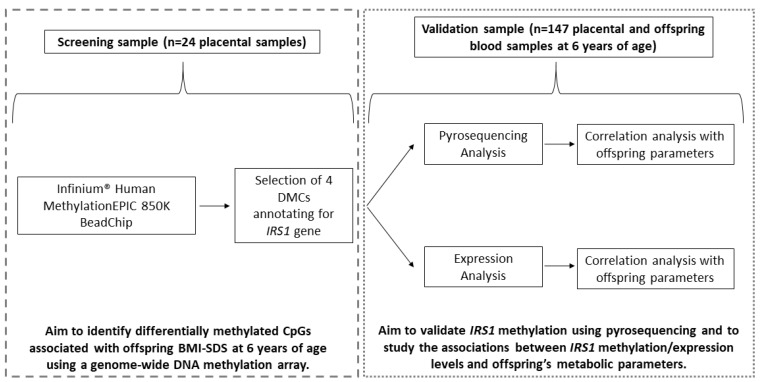
Flow chart with the study design.

**Table 1 ijms-26-03141-t001:** Clinical characteristics of the study subjects.

	Screening	Validation	*p*-Value
**Mother (n)**	**24**	**147**	
Age (yr)	31 ± 1	31 ± 1	NS
Height (cm)	164 ± 1	163 ± 1	NS
Pregestational weight (kg)	68.5 ± 2.9	65.6 ± 1.0	NS
Pregestational BMI (kg/m^2^)	25.2 ± 1.0	24.7 ± 0.3	NS
Pregestational obesity (%)	30	34	NS
**Newborn (n)**	**24**	**147**	
Gender (% female)	50	52	NS
Gestational age (wk)	40 ± 0.1	40 ± 0.1	NS
Birth weight (kg)	3.4 ± 0.1	3.4 ± 0.1	NS
Birth weight-SDS	0.3 ± 0.1	0.2 ± 0.1	NS
Birth length (cm)	50.1 ± 0.2	49.7 ± 0.1	NS
Birth length-SDS	0.07 ± 0.1	0.01 ± 0.1	NS
**Offspring at 6 yr (n)**	**24**	**147**	
Age (yr)	6.2 ± 0.1	6.0 ± 0.1	NS
Weight (kg)	23.7 ± 1.0	22.4 ± 0.4	NS
Weight-SDS	0.22 ± 0.2	0.05 ± 0.1	NS
Height (cm)	120 ± 1	116 ± 1	NS
Height-SDS	0.58 ± 0.2	0.11 ± 0.1	NS
BMI (kg/m^2^)	16.3 ± 0.3	16.3 ± 0.1	NS
BMI-SDS	−0.02 ± 0.1	0.01 ± 0.1	NS
Δ BW-SDS to weight-SDS	−0.18 ± 0.2	−0.15 ± 0.1	NS
Waist (cm)	57.1 ± 1.7	56.3 ± 0.6	NS
Hip (cm)	61.1 ± 1.8	59.6 ± 0.6	NS
SBP (mmHg)	96.9 ± 3.0	95.9 ± 1.0	NS
DBP (mmHg)	57.1 ± 1.1	57.0 ± 0.7	NS
HDL-cholesterol (mg/dL)	57.0 ± 2.7	55.8 ± 0.8	NS
Triglycerides (mg/dL)	49.5 ± 2.7	50.8 ± 1.2	NS
Glucose (mg/dL)	85.0 ± 1.7	82.9 ± 0.5	NS
Insulin (mIU/L)	6.2 ± 0.5	5.2 ± 0.2	NS
HOMA-IR	1.3 ± 0.1	1.1 ± 0.1	NS
FBM (kg)	5.9 ± 0.6	5.6 ± 0.2	NS
LBM (kg)	18.0 ± 0.6	17.0 ± 0.2	NS
Subcutaneous fat (cm)	0.41 ± 0.03	0.45 ± 0.02	NS
Peritoneal fat (cm)	0.45 ± 0.04	0.46 ± 0.01	NS
Visceral fat (cm^2^)	5.4 ± 0.2	5.2 ± 0.1	NS

Data are shown as mean ± SEM. BMI: body mass index; SDS: standard deviation score; ∆ BW-SDS to weight-SDS: z-score changes from weight at birth to weight at 6 years; SBP: systolic blood pressure; DBP: diastolic blood pressure; HDL: high-density lipoprotein; HOMA-IR: homeostatic model assessment for insulin resistance; FBM: fat body mass; LBM: lean body mass; NS: non-significant.

**Table 2 ijms-26-03141-t002:** Compilation of the top hypermethylated (**A**) and hypomethylated (**B**) DMCs.

(A) Hypermethylated DMCs						
Ilmn ID	Beta Coef.	FDR	OR	Chr	Position	Gene
cg00406870	1.03970511	4.35 × 10^−43^	2.82838283	11	124981741	**TMEM218**
cg07150062	1.03245317	2.74 × 10^−39^	2.80794576	14	104552032	**ASPG**
cg10761315	0.98664557	7.57 × 10^−37^	2.68222203	14	104552034	**ASPG**
cg11163620	0.97173534	5.92 × 10^−36^	2.64252615	4	157997554	GLRB
cg01963620	0.97115961	6.50 × 10^−35^	2.64100522	11	124981674	**TMEM218**
cg08626939	0.96229948	4.80 × 10^−33^	2.61770892	2	227656417	**IRS1**
cg05665562	0.96004491	5.03 × 10^−38^	2.61181376	11	124981679	**TMEM218**
cg14874299	0.95164064	4.12 × 10^−36^	2.58995536	11	124981343	**TMEM218**
cg12163935	0.94020019	8.88 × 10^−32^	2.56049395	2	227656057	**IRS1**
cg05446424	0.92323155	3.13 × 10^−43^	2.51741242	2	14772734	FAM84A
**(B) Hypomethylated DMCs**						
**Ilmn ID**	**Beta Coef.**	**FDR**	**OR**	**Chr**	**Position**	**Gene**
cg10324224	−1.13770315	6.06 × 10^−11^	0.32055444	1	231115997	TTC13
cg24202000	−1.09504286	2.49 × 10^−10^	0.33452527	8	129551766	LINC00824
cg21240123	−0.98681175	1.42 × 10^−08^	0.37276326	3	20016987	RAB5A
cg14730097	−0.96058097	8.10 × 10^−09^	0.3826705	2	233632281	GIGYF2
cg18705155	−0.86117235	7.25 × 10^−09^	0.42266628	6	39194077	KCNK5
cg20401473	−0.81442859	7.13 × 10^−07^	0.44289233	6	88186912	SLC35A1
cg23691406	−0.80357486	5.41 × 10^−17^	0.44772554	14	71112909	TTC9
cg10533159	−0.77469288	8.47 × 10^−40^	0.4608453	1	207991937	LOC148696
cg17325094	−0.77392864	2.08 × 10^−20^	0.46119763	1	57809419	DAB1
cg04248557	−0.74101842	1.63 × 10^−06^	0.47662826	7	69196758	AUTS2

In bold, we indicate DMCs annotated to the same gene, aligned in the same direction, and located close to each other. FDR, false discovery rate-adjusted *p*-value; OR, odds ratio; Chr, chromosome.

**Table 3 ijms-26-03141-t003:** Correlations between placental *IRS1* (CpG2) methylation and expression and parameters related to body composition and metabolic risk in children at 6 years of age.

	Placental *IRS1* (CpG2) Methylation						Placental *IRS1* Expression					
Offspring at 6 yr	All		Boys		Girls		All		Boys		Girls	
	r	*p*-Value	r	*p*-Value	r	*p*-Value	r	*p*-Value	r	*p*-Value	r	*p*-Value
Weight-SDS	**0.227**	**0.006**	**0.254**	**0.03**	**0.249**	**0.02**	0.054	Ns	**0.319**	**0.006**	−0.011	NS
Height-SDS	**0.160**	**0.05**	0.214	NS	0.142	NS	0.134	NS	0.150	NS	0.091	NS
BMI-SDS	**0.221**	**0.007**	**0.245**	**0.04**	**0.237**	**0.03**	0.062	NS	0.118	NS	−0.016	NS
Δ BW-SDS to weight-SDS	**0.190**	**0.02**	**0.236**	**0.05**	0.201	NS	0.039	NS	0.153	NS	−0.063	NS
Waist	**0.215**	**0.01**	**0.235**	**0.05**	**0.225**	**0.05**	0.092	NS	0.168	NS	0.079	NS
Hip	**0.217**	**0.01**	**0.270**	**0.03**	0.163	NS	0.149	NS	0.287	0.02	0.030	NS
Waist-to-height ratio	**0.181**	**0.03**	0.155	NS	0.214	NS	0.098	NS	0.091	NS	−0.115	NS
LBM	**0.185**	**0.03**	**0.240**	**0.05**	0.201	NS	0.168	0.05	0.317	0.01	0.123	NS
FBM	**0.176**	**0.04**	0.195	NS	0.164	NS	0.078	NS	0.198	NS	−0.029	NS
Subcutaneous fat	**0.248**	**0.003**	**0.270**	**0.02**	**0.244**	**0.03**	0.045	NS	0.086	NS	−0.137	NS
Preperitoneal fat	**0.171**	**0.03**	**0.282**	**0.01**	0.066	NS	0.086	NS	0.032	NS	−0.182	NS
Visceral fat	0.143	NS	**0.242**	**0.04**	0.063	NS	**0.248**	**0.003**	**0.408**	**<0.0001**	0.057	NS
Insulin	0.103	NS	−0.041	NS	**0.241**	**0.03**	0.167	0.05	0.178	NS	**0.249**	**0.03**
HOMA-IR	0.063	NS	−0.086	NS	0.204	NS	0.159	0.05	0.179	NS	**0.264**	**0.02**

In bold, we indicate values that are significant after adjusting for maternal BMI and offspring age and gender in the whole population and only for the maternal BMI in boys and girls. SDS: standard deviation score; BMI: body mass index; ∆ BW-SDS to weight-SDS: z-score changes from weight at birth to weight at 6 years; LBM: lean body mass; FBM: fat body mass; HOMA-IR: homeostatic model assessment for insulin resistance; NS: non-significant.

**Table 4 ijms-26-03141-t004:** Correlations between leukocyte *IRS1* expression and parameters related to body composition and metabolic risk in children at 6 years of age.

	Offspring Leukocyte *IRS1* Expression					
Offspring at 6yr	All		Boys		Girls	
	r	*p*-Value	r	*p*-Value	r	*p*-Value
Weight-SDS	0.183	NS	−0.074	NS	**0.470**	**0.001**
Height-SDS	0.094	NS	0.012	NS	0.186	NS
BMI-SDS	0.148	NS	−0.152	NS	**0.444**	**0.002**
Δ BW-SDS to weight-SDS	0.028	NS	−0.177	NS	0.283	0.05
Waist	0.120	NS	−0.230	NS	**0.460**	**0.002**
Hip	0.025	NS	−0.207	NS	0.258	NS
Waist-to-height ratio	**0.214**	**0.05**	−0.157	NS	**0.470**	**0.001**
LBM	0.043	NS	−0.109	NS	0.252	NS
FBM	0.084	NS	−0.176	NS	**0.431**	**0.004**
Subcutaneous fat	0.107	NS	−0.091	NS	**0.313**	**0.03**
Preperitoneal fat	0.100	NS	0.007	NS	0.178	NS
Visceral fat	**0.226**	**0.03**	−0.099	NS	**0.411**	**0.006**
Insulin	0.080	NS	−0.155	NS	**0.383**	**0.009**
HOMA-IR	0.066	NS	−0.144	NS	**0.335**	**0.02**

In bold, we indicate values that were significant after adjusting for maternal BMI and offspring age and gender in the whole population and only for the maternal BMI in boys and girls. SDS: standard deviation score; BMI: body mass index; ∆ BW-SDS to weight-SDS: z-score changes from weight at birth to weight at 6 years; LBM: lean body mass; FBM: fat body mass; HOMA-IR: homeostatic model assessment for insulin resistance; NS: non-significant.

## Data Availability

The DNA methylation data analyzed during the current study are publicly available and have been deposited in the Gene Expression Omnibus data repository under accession number GSE192812. Other data described in the manuscript, code book, and analytic code will be made available upon request to the corresponding author (J.B.).

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
