# Peer review of "Longitudinal Analysis of Placental IRS1 DNA Methylation and Childhood Obesity"

_ijms, 2025, doi:10.3390/ijms26073141_

Round 1
Reviewer 1 Report
Comments and Suggestions for Authors
Dear Author,
The Manuscript No. ijms-3544857, Longitudinal analysis of placental IRS1 DNA methylation and childhood obesity, presents an important contribution to the field of nutrition and metabolic research, particularly regarding the role of DNA methylation in placental tissue and its association with childhood obesity. The study is well-structured, employs robust statistical methodologies, and presents relevant findings that could advance knowledge in this area. However, several aspects require revision:
Consider using person-first language as this is the standard when discussing individuals with chronic conditions like obesity.
The Supplementary Methods mention that only offspring age and sex were considered as confounding factors. However, other important variables, such as maternal diet, socioeconomic status, and paternal obesity, were not included.
It would be helpful to explain why these weren’t considered and whether their absence might affect the results.
Since the study focuses on a Caucasian population, it would be great to discuss how the findings might apply to other ethnic groups and what limitations might exist in terms of broader applicability.
Some abbreviations in Tables 3 and 4, and Figure 3 aren’t defined. Add a footnote or legend explaining them
References: There are some very old references (from 1994, 2001, 2008, 2010, 2011, 2012, 2013, 2017) can you replace them with more updated ones?
The authors should revise the English to remove redundant sentences, improve conciseness, and adjust punctuation as needed
Comments on the Quality of English Language
The authors should revise the English to remove redundant sentences, improve conciseness, and adjust punctuation as needed
Reviewer 2 Report
Comments and Suggestions for Authors
This MS titled “Longitudinal analysis of placental IRS1 DNA methylation and childhood obesity. The study employs a genome-wide DNA methylation analysis (EWAS)”, explores the role of IRS1 DNA methylation in placental and blood samples as a potential biomarker for childhood obesity. The study employs EWAS, followed by pyrosequencing validation and gene expression assessment, LRM and ML to evaluate predictive associations. This is a well-structured study that addresses an important question. The authors successfully identify differentially methylated CpG sites in IRS1, validate them in cohort, and assess their functional significance. The MS is nicely written, and the methodology is sounds, however, some areas require clarification and refinement.
Major
- The paper hints that IRS1 is a biomarker for predicting obesity and metabolic risk but does not clarify whether it was validated in independent cohorts. The authors should explicitly state whether IRS1 methylation has been replicated in other datasets. If it has not been externally validated, they should acknowledge this limitation. Additionally, the manuscript should provide a brief rationale, in the introduction, why IRS1 is considered a key regulator in obesity and metabolism, along with an explanation of why it is hypothesized to be epigenetically modified in placental tissue.
- The statement below is not referenced in the introduction and requires citations. "Several findings from the aforementioned studies suggest that childhood overweight and obesity may result from alterations in the epigenome. Conversely, other studies propose that obesity itself may influence the epigenome."
- The statement below is strong, please keep in mind that while some epigenetic marks are stable, others are highly dynamic and tissue-specific, influenced by other factors such as diet, environment, and growth. “We further propose that some of these methylation marks may persist over time and could serve as early predictors of childhood obesity and metabolic risk”.
- The MS states in objective 2 “to assess whether these DNA methylation marks can be detected in blood samples at age 6 years and are also associated with obesity-related outcomes at that time." However, it is unclear whether the authors expect a direct correlation between placental and blood methylation patterns. Given that placental and blood methylation do not always align as some CpG sites exhibit tissue-specific methylation this should be acknowledged.
- The study assumes that placental methylation patterns persist into childhood and can be detected in blood. However, it is important to note that methylation signatures are often tissue-specific, and placental epigenetic marks do not always reflect systemic methylation changes. The authors should consider acknowledging this as a limitation of the study and discussing the potential implications of tissue-specific variability in DNA methylation patterns.
Minor
- Since IJMS typically follows American English conventions, it is recommended to change "leucocyte" to "leukocyte" for consistency.
- In the abstract, considered re-phrase “to be also associated with obesity-related outcomes” to “whether they correlate with obesity-related outcomes”.
